# AIGID-RFT: Reinforcement Fine-Tuning Multimodal LLMs for AI-Generated Image Detection

## Abstract

The rapid progress of generative artificial intelligence has made AI-generated image (AIGI) detection increasingly critical for digital forensics and trustworthy media. Existing AIGI detectors are effective on raw images but lack robustness against post-processing operations. Meanwhile, multimodal large language models (MLLMs) have demonstrated strong general capabilities, but their direct application to AIGI detection remains limited. To address these challenges, we propose **AIGID-RFT**, a novel MLLM-based AI-generated image detector. Unlike prior methods that rely on supervised fine-tuning, we adopt reinforcement learning as the post-training paradigm and design verifiable rewards tailored for the AIGI detection task, thereby unlocking the intrinsic potential of MLLMs. Furthermore, we carefully design a Cross Layer Forensic Adapter, which is integrated in parallel with the vision encoder to effectively exploit multi-level visual features for enhanced detection performance. Our method requires only binary labels for training, eliminating the need for costly text annotations. Extensive experiments demonstrate that our method significantly outperforms existing AIGI detectors under diverse post-processing operations that simulate real-world scenarios.

## 1 Introduction

In recent years, the rapid development of generative AI (Podell et al., 2023; Esser et al., 2024) has enabled the generation of high-quality images, which have substantially enriched the content on social media. However, these realistic AI-generated images can also facilitate misinformation and opinion manipulation (Xu et al., 2023), which has raised widespread concerns about the authenticity of visual content. Therefore, developing effective and robust detectors for AI-generated images is critical to mitigating their potential misuse.

When images spread on social platforms, they often undergo varying degrees of compression and resizing. Consequently, robustness to post-processing operations is critical for ensuring the reliability of AI-generated image detectors in real-world scenarios. Some previous methods achieve near-perfect performance on original images but are insufficiently evaluated under image degradation. We adopt the latest AIGIBench (Li et al., 2025c) as the primary evaluation benchmark. It covers 25 sources of AI-generated images and provides comprehensive evaluation under various post-processing operations. We evaluate the robustness of our proposed method and 11 state-of-the-art detectors on AIGIBench, with a detailed analysis provided in Section 2. Experiments show that under JPEG compression with a quality factor of 50, which simulates real-world scenarios, existing detectors suffer a sharp performance drop, particularly in recall for AI-generated images. These results highlight a major limitation of existing methods: insufficient robustness to JPEG compression.

Multimodal large language models (MLLMs) demonstrate strong capabilities across various domains and hold the potential for providing language-based explanations in image forensics tasks (Zou et al., 2025). The LOKI benchmark (Ye et al., 2025) evaluates the performance of different MLLMs on AIGC detection, but results show that directly applying MLLMs for detection performs poorly, as they lack effective recognition of AI-generated images. Reinforcement learning (RL) as a post-training paradigm for LLMs has shown considerable potential. In particular, Group Relative Policy Optimization (GRPO) (Shao et al., 2024), a reinforcement learning with verifiable

rewards (RLVR) method, computes rewards directly from model outputs, eliminating the need for costly text annotations. Subsequent works (Chen et al., 2025; Liu et al., 2025) have extended GRPO to multimodal LLMs (MLLMs), showing generalization capabilities on tasks such as image classification and object detection. However, RLVR-based MLLM post-training remains unexplored for AI-generated image detection.

To address the robustness limitations of prior AIGI detectors and unlock the intrinsic potential of MLLMs for AIGI detection, we propose AIGID-RFT, a novel MLLM-based AI-generated image detector. Unlike previous methods that rely on supervised fine-tuning to provide MLLMs with forensic capabilities (Li et al., 2025b; Wen et al., 2025; Zhou et al., 2025), we adopt reinforcement learning as the post-training paradigm and design verifiable rewards tailored for the AIGI detection task, thereby eliminating the dependence on costly text annotations. Furthermore, inspired by prior works that enhance MLLMs using intermediate visual features (Cao et al., 2024; Yao et al., 2024), we design a Cross Layer Forensic Adapter (CLFA), which is integrated in parallel with multiple intermediate layers of the MLLM vision encoder. Multiple CLFAs share the same parameters and process visual features from different intermediate layers separately. Their outputs are then added back to the final visual feature of the encoder. This design enables the MLLM to fully leverage multi-level visual features, thereby improving its performance in detecting AI-generated images.

Our main contributions are summarized as follows:

- We propose a novel method, **AIGID-RFT**, which utilizes reinforcement learning with verifiable rewards to guide MLLMs to effectively detect AI-generated images. Our method relies only on "real/fake" labels for training, without requiring extensive textual annotations.

- We design the Cross Layer Forensic Adapter (CLFA) and integrate it in parallel into multiple intermediate layers of the visual encoder, utilizing multi-level visual features to improve detection performance.

- Extensive experiments demonstrate the effectiveness of our method for AIGI detection and show that it significantly outperforms existing detectors under diverse post-processing conditions simulating real-world scenarios.

## 2 RETHINKING THE ROBUSTNESS OF AI-GENERATED IMAGE DETECTORS

In this section, we systematically analyze existing AI-generated image detectors and identify a key limitation: insufficient robustness to JPEG compression. We adopt the recently released AI-GIBench (Li et al., 2025c) as the evaluation benchmark and evaluate the robustness of our proposed AIGID-RFT along with 11 state-of-the-art detection methods. Further details on the dataset and baselines are provided in Section 4.1.

These existing AIGI detectors typically train a binary classifier to distinguish between natural and AI-generated images, which can be broadly categorized into two types: (1) CLIP-based detectors (Ojha et al., 2023; Yan et al., 2025c) train a linear classifier on image embeddings extracted by the CLIP visual encoder. (2) Artifact-based detectors focus on identifying characteristic artifacts left by generators in images. For example, NPR (Tan et al., 2024b) captures artifacts caused by upsampling operations in generators, while DFFreq (Yan et al., 2025a) and SAFE (Li et al., 2025a) extract synthetic artifacts through frequency-domain analysis. Although these methods focu on different types of artifacts, their core idea is to utilize low-level image signals for detection.

Figure 1 shows the average fake accuracy on AIGIBench, illustrating the impact of JPEG compression on the performance of different AI-generated image detectors. The underlined methods rely on low-level artifacts or texture features, making them highly sensitive to JPEG compression. Under JPEG compression with a quality factor of 50, the fake accuracy of these artifact-based detectors drops sharply, nearly to 0%, because compression significantly weakens the low-level artifacts left by generative models. In contrast, CLIP-based detectors, such as CLIPD (Ojha et al., 2023) and Effort (Yan et al., 2025c), utilize CLIP image embeddings and maintain reasonable detection performance under JPEG compression. Based on this observation, our method addresses the robustness issue by employing an MLLM as the base model, whose semantic-level representations offer stronger resistance to post-processing. Notably, our proposed AIGID-RFT achieves better robustness than previous methods under the same conditions.

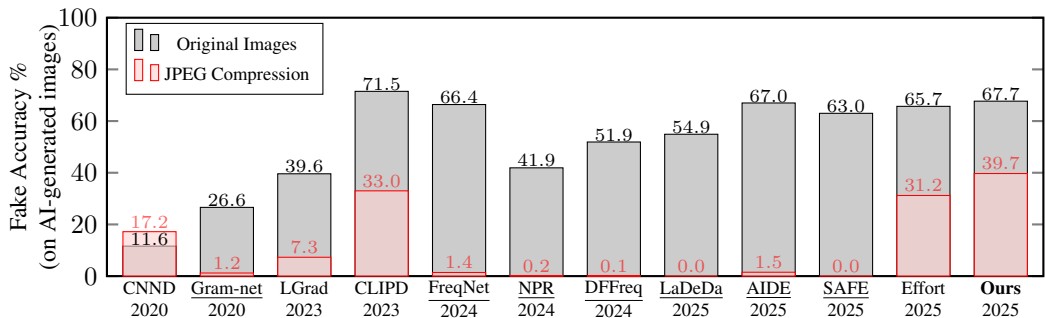

Figure 1: Robustness of AI-Generated Image Detectors under JPEG Compression. We report the mean fake accuracy (i.e., the recall on AI-generated images) across all subsets on AIGIBench. Gray bars represent original images, while pink bars represent JPEG-compressed images (quality = 50). An underline indicates that the method involves low-level artifact or texture analysis.

## 3 METHOD

In this section, we propose AIGID-RFT, a novel method for detecting AI-generated images. As shown in Figure 2, our model utilizes Qwen2.5-VL-7B-Instruct (Team, 2025) as the base model. We further design the Cross Layer Forensic Adapter (CLFA) and integrate it in parallel into different intermediate layers of the visual encoder, enabling the LLM to utilize multi-level visual features and fully exploit the potential of MLLMs for AIGI detection. In the following subsections, we provide a detailed description of each component of the model, as well as its reinforcement training and inference phases.

### 3.1 OVERALL ARCHITECTURE OF AIGID-RFT

Our model follows the standard paradigm of multimodal large language models (MLLMs), "ViT-MLP-LLM", and consists of a visual encoder, an MLP projector, and a large language model. The visual encoder adopts a Vision Transformer (ViT) (Dosovitskiy et al., 2020) architecture. The height and width of the input image $x$ are first resized to multiples of 28, then divided into $14 \times 14$ patches and processed by the visual encoder to obtain the image features:

$$\mathbf{F}_{\text{img}} = E_{\text{v}}(x), \tag{1}$$

where $E_{\text{v}}$ denotes the ViT visual encoder. The visual features extracted by the ViT are mapped through the projector to align with the LLM text embedding dimensions, while the question text $q$ is encoded by a tokenizer:

$$\mathbf{T}_{\text{img}} = E_{\text{p}}(\mathbf{F}_{\text{img}}), \quad \mathbf{T}_{\text{text}} = E_{\text{t}}(q), \tag{2}$$

where $E_{\text{p}}$ denotes the projector and $E_{\text{t}}$ denotes the tokenizer. The visual tokens and text tokens are then fed into the LLM to generate the final textual output:

$$o = \text{LLM}(\mathbf{T}_{\text{img}}, \mathbf{T}_{\text{text}}), \tag{3}$$

where LLM denotes the large language model, and $o$ is its output, which contains both the reasoning process for the input image and the final authenticity classification "real/fake".

### 3.2 CROSS LAYER FORENSIC ADAPTER

We design the Cross Layer Forensic Adapter (CLFA) and integrate it in parallel into the intermediate layers of the visual encoder, enabling the model to leverage multi-level visual features and improve its performance in AI-generated image detection tasks. Specifically, the visual encoder of Qwen2.5-VL can be divided into four blocks, each containing several window attention layers and one full attention layer. We denote the output of each block as $\mathbf{F}_{\text{img}}^i$, $i = 1, \ldots, 4$, where $\mathbf{F}_{\text{img}}^4$ corresponds to the original final output features of visual encoder. We integrate the CLFA in parallel across different blocks of the visual encoder. Concretely, the output features of the intermediate blocks are fed into

Figure 2: Overall architecture of AIGID-RFT. We adopt Qwen2.5-VL-7B as the base model, and we introduce a Cross Layer Forensic Adapters (CLFA) which is integrated in parallel with the ViT backbone to fully leverage multi-level visual features. The entire model is trained end-to-end with GRPO, aiming to fully exploit its potential for AIGI detection. During inference, we compute the probabilities of real and fake based on the logits of the corresponding tokens.

these shared-parameter CLFAs, and their outputs are added to the final visual encoder features to obtain the enhanced image features:

$$\mathbf{F}_{\text{img}} = \mathbf{F}_{\text{img}}^4 + \sum_{i=1}^{3} \text{CLFA}(\mathbf{F}_{\text{img}}^i), \tag{4}$$

where CLFA denotes our proposed Cross Layer Forensic Adapter, implemented as an MLP network containing a LayerNorm (Zhang & Sennrich, 2019) followed by two linear layers. The obtained $\mathbf{F}_{\text{img}}$ contains multi-level features from the visual encoder, enabling more comprehensive capture of patterns specific to AI-generated images.

Previous works that utilize intermediate features from visual encoders (Cao et al., 2024; Yao et al., 2024) require modifications to the model architecture and additional pretraining, whereas our CLFA is integrated as an adapter on the original model, enabling direct end-to-end reinforcement learning.

## 3.3 TRAINING PHASE

We adopt Group Relative Policy Optimization (GRPO) (Shao et al., 2024) as our reinforcement learning strategy and design verifiable rewards for the AI-generated image detection task to train our model end-to-end.

**Group Relative Policy Optimization** For each question $q$, GRPO samples $G$ outputs $\{o_1, o_2, \ldots, o_G\}$ from the old policy $\pi_{\theta_{old}}$, and optimizes the policy model by maximizing the following objective:

$$\mathcal{J}(\theta) = \mathbb{E}[q \sim P(Q), \{o_i\}_{i=1}^G \sim \pi_{\theta_{old}}(O|q)]$$

$$\frac{1}{G} \sum_{i=1}^{G} \frac{1}{|o_i|} \sum_{t=1}^{|o_i|} \left\{ \min \left[ w_{i,t}(\theta)\hat{A}_{i,t}, \text{clip}\left(w_{i,t}(\theta), 1-\epsilon, 1+\epsilon\right)\hat{A}_{i,t} \right] - \beta \mathbb{D}_{KL}\left[\pi_\theta || \pi_{ref}\right] \right\}, \tag{5}$$

where $\epsilon$ controls the clipping range, $\beta$ is the coefficient of the KL-divergence penalty, $w_{i,t}(\theta)$ is the importance sampling ratio, and $\hat{A}_{i,t}$ is the relative advantage computed from the reward function. $w_{i,t}(\theta)$ and $\hat{A}_{i,t}$ are defined as:

$$w_{i,t}(\theta) = \frac{\pi_\theta(o_{i,t}|q, o_{i,<t})}{\pi_{\theta_{old}}(o_{i,t}|q, o_{i,<t})}, \quad \hat{A}_{i,t} = \frac{r(q, o_i) - \text{mean}\left(\{r(q, o_i)\}_{i=1}^G\right)}{\text{std}\left(\{r(q, o_i)\}_{i=1}^G\right)}, \tag{6}$$

where $r(q, o_i)$ denotes the reward function, which includes a format reward and an accuracy reward.

**Format Reward** The format reward encourages the model to follow a structured reasoning process. It guides the model to output its reasoning process within `<think>` and `</think>` tags, and to include the final authenticity judgment "real/fake" within `<answer>` and `</answer>` tags. If the model outputs the correct format, the format reward $r_{\text{format}}$ is set to 1; otherwise, it is 0.

**Accuracy Reward** To support AI-generated image detection, we adopt a detection reward based on consistency between the predicted and ground truth classes. Specifically, if the class predicted by the model between `<answer>` and `</answer>` matches the ground truth, the accuracy reward $r_{\text{accuracy}}$ is set to 1; otherwise, it is 0.

The final reward function is defined as:

$$r(q, o) = r_{\text{format}} + r_{\text{accuracy}}. \tag{7}$$

Our method uses verifiable rewards, requiring only images and labels as training data, thereby eliminating the need for extensive textual annotations required by SFT.

### 3.4 INFERENCE PHASE

During inference, we take the class predicted by the model between `<answer>` and `</answer>` as the final prediction and use it to compute accuracy. To further obtain the model's classification confidence, we extract the output logits for the "real" and "fake" label tokens at this position, denoted as $z_{\text{real}}$ and $z_{\text{fake}}$. We then apply the softmax function to these two logits to obtain normalized classification probabilities, where the predicted probability for the "fake" label is defined as:

$$p_{\text{fake}} = \frac{\exp(z_{\text{fake}})}{\exp(z_{\text{real}}) + \exp(z_{\text{fake}})}, \tag{8}$$

where $\exp(\cdot)$ denotes the exponential function $e^x$. The probability $p_{\text{fake}}$ serves as the confidence score for computing the Average Precision (A.P.) in subsequent evaluations.

## 4 EXPERIMENT

### 4.1 EXPERIMENTAL SETUP

**Benchmark and Baselines.** We compare our method with 11 state-of-the-art AI-generated image detection methods on the recently proposed AIGIBench (Li et al., 2025c). AIGIBench contains 25 subsets, covering AI-generated images from different generators as well as images generated on social media platforms. This benchmark provides a comprehensive evaluation of our method's generalization to unseen generators and robustness under various image post-processing conditions. The compared detectors include: CNND (Wang et al., 2020), Gram-Net (Liu et al., 2020), LGrad (Tan et al., 2023), CLIPD (Ojha et al., 2023), FreqNet (Tan et al., 2024a), NPR (Tan et al., 2024b), LaDeDa (Cavia et al., 2024), DFFreq (Yan et al., 2025a), AIDE (Yan et al., 2025b), SAFE (Li et al., 2025a), CO-SPY Cheng et al. (2025), and Effort (Yan et al., 2025c). We retrain CO-SPY Cheng et al. (2025) and Effort (Yan et al., 2025c) on the AIGIBench training set using their official code, while results for the other methods are cited from the AIGIBench paper. Note that the two retrained methods and our method do not use any data augmentation during the training stage. In addition, we evaluate on the image judgement task of the LOKI Benchmark (Ye et al., 2025) and compare with 20 multimodal large language models (MLLMs). We additionally test Qwen2.5-VL-7B, while results for other models are cited from the original LOKI paper.

**Metrics.** Following prior work and AIGIBench (Li et al., 2025c), we use classification accuracy (Acc.) and average precision (A.P.) as our primary evaluation metrics. In addition, we decompose accuracy into two complementary components: R.Acc and F.Acc, which correspond to the detector's accuracy on real images and AI-generated images, respectively. These metrics provide a more detailed and informative assessment of the detector's effectiveness.

Table 1: Comparison with state-of-the-art AI-generated image detectors on AIGIbench. The table reports the Average Precision (A.P.) for each of the 25 subsets, and the mean A.P. across all subsets. All methods are trained on the same dataset containing images from SD-v1.4 and ProGAN. The best and the second-best mean performance are indicated by **bold** and underline, respectively.

| Method Year | CNND 2020 | Gram-net 2020 | LGrad 2023 | CLIPD 2023 | FreqNet 2024 | NPR 2024 | DFFreq 2024 | LaDeDa 2025 | AIDE 2025 | SAFE 2025 | CO-SPY 2025 | Effort 2025 | **Ours** 2025 |
|---|---|---|---|---|---|---|---|---|---|---|---|---|---|
| *Diffusion for Text-to-Image Generation* | | | | | | | | | | | | | |
| FLUX1-dev | 72.3 | 75.1 | 88.1 | 79.5 | 87.3 | 99.0 | 86.3 | 98.7 | 93.4 | 99.7 | 87.3 | 81.2 | 96.6 |
| Midjourney-V6 | 59.8 | 41.6 | 64.5 | 61.5 | 55.9 | 76.9 | 68.1 | 86.9 | 83.0 | 98.4 | 74.6 | 77.4 | 94.6 |
| GLIDE | 60.0 | 84.3 | 91.0 | 80.3 | 77.4 | 94.3 | 96.3 | 95.0 | 97.7 | 97.9 | 84.6 | 93.5 | 83.9 |
| DALLE-3 | 68.6 | 61.1 | 62.7 | 76.3 | 61.0 | 70.0 | 58.6 | 59.8 | 63.1 | 45.8 | 85.6 | 82.7 | 99.6 |
| Imagen3 | 57.4 | 54.7 | 69.7 | 79.3 | 80.7 | 94.4 | 87.3 | 97.2 | 95.2 | 98.8 | 76.9 | 74.7 | 98.4 |
| SD3 | 73.1 | 60.0 | 72.1 | 87.2 | 82.6 | 97.2 | 90.8 | 98.7 | 98.3 | 98.8 | 95.6 | 91.9 | 98.3 |
| SDXL | 64.2 | 76.9 | 82.8 | 88.0 | 95.2 | 94.4 | 95.8 | 98.5 | 95.7 | 99.7 | 92.6 | 91.9 | 99.9 |
| BLIP | 92.9 | 99.9 | 97.4 | 95.8 | 100.0 | 100.0 | 99.6 | 99.9 | 95.5 | 100.0 | 95.5 | 99.9 | 98.1 |
| *Diffusion for Personalized Generation* | | | | | | | | | | | | | |
| Infinite-ID | 49.5 | 60.1 | 54.6 | 89.6 | 74.5 | 80.4 | 82.7 | 76.9 | 94.7 | 99.2 | 97.4 | 98.7 | 99.8 |
| InstantID | 80.2 | 85.7 | 81.5 | 93.5 | 86.3 | 79.2 | 97.2 | 90.4 | 96.3 | 99.6 | 96.7 | 99.0 | 99.2 |
| IP-Adapter | 65.8 | 64.2 | 78.3 | 87.3 | 79.9 | 91.7 | 91.3 | 94.3 | 95.4 | 98.1 | 84.3 | 91.0 | 95.9 |
| PhotoMaker | 58.2 | 58.6 | 67.2 | 72.3 | 74.9 | 43.6 | 94.0 | 90.7 | 95.6 | 99.3 | 57.3 | 82.3 | 90.8 |
| *GAN-based Noise-to-Image Generation* | | | | | | | | | | | | | |
| ProGAN | 99.9 | 100.0 | 99.8 | 99.9 | 100.0 | 100.0 | 100.0 | 100.0 | 99.6 | 100.0 | 99.9 | 100.0 | 97.3 |
| R3GAN | 52.7 | 52.5 | 58.7 | 91.2 | 56.8 | 61.1 | 74.0 | 72.6 | 97.1 | 98.2 | 95.9 | 95.8 | 73.6 |
| StyleGAN3 | 73.1 | 77.9 | 80.5 | 84.5 | 92.4 | 91.7 | 96.4 | 96.9 | 91.4 | 97.6 | 91.3 | 94.9 | 98.7 |
| StyleGAN-XL | 64.2 | 83.7 | 74.6 | 93.3 | 84.1 | 75.3 | 83.4 | 98.5 | 93.2 | 97.6 | 96.1 | 93.7 | 81.4 |
| StyleSwim | 76.5 | 86.0 | 90.0 | 95.2 | 91.8 | 94.9 | 92.9 | 98.5 | 89.3 | 99.6 | 93.4 | 99.7 | 91.8 |
| WFIR | 50.0 | 43.9 | 49.4 | 82.0 | 48.9 | 65.5 | 82.2 | 86.9 | 90.8 | 81.8 | 87.4 | 93.8 | 90.2 |
| *GANs for Deepfake* | | | | | | | | | | | | | |
| BlendFace | 73.4 | 33.3 | 34.9 | 35.3 | 34.1 | 34.7 | 35.2 | 42.1 | 54.2 | 45.6 | 41.7 | 40.4 | 44.4 |
| E4S | 68.9 | 32.5 | 32.8 | 57.1 | 34.7 | 34.4 | 34.8 | 49.3 | 44.3 | 46.0 | 67.4 | 56.7 | 64.1 |
| FaceSwap | 58.7 | 34.6 | 37.5 | 52.4 | 43.4 | 43.6 | 45.6 | 40.9 | 56.3 | 45.7 | 50.3 | 59.9 | 58.2 |
| InSwap | 77.9 | 38.0 | 35.0 | 40.2 | 42.1 | 40.7 | 41.2 | 47.4 | 54.6 | 49.9 | 41.4 | 51.3 | 50.0 |
| SimSwap | 70.0 | 36.4 | 37.6 | 40.4 | 41.9 | 42.7 | 43.8 | 42.3 | 62.7 | 49.5 | 41.0 | 58.4 | 57.5 |
| *Open-source Platforms* | | | | | | | | | | | | | |
| SocialRF | 50.6 | 53.0 | 54.9 | 55.2 | 58.1 | 68.4 | 63.3 | 68.3 | 65.0 | 64.2 | 72.9 | 57.1 | 98.9 |
| CommunityAI | 59.1 | 66.1 | 69.4 | 73.2 | 69.7 | 62.9 | 52.1 | 56.3 | 61.0 | 55.2 | 69.3 | 60.1 | 87.0 |
| **Mean A.P. (%)** | 67.1 | 62.4 | 66.6 | 75.6 | 70.1 | 73.9 | 75.7 | 79.3 | 82.7 | 82.6 | 79.1 | 81.0 | **85.9** |

**Implementation Details.** We use Qwen2.5-VL-7B-Instruct (Team, 2025) as the base model. To construct the GRPO training set, we randomly sample 2,000 images from the official training set of AIGIBench, which consists of images generated by SDv1.4 (Rombach et al., 2022) and Pro-GAN (Karras et al., 2018). Our proposed CLFA contains an RMSNorm (Zhang & Sennrich, 2019) and two linear layers of dimensions (1280, 80) and (80, 1280), and it is integrated into layers indexed by $\{7,15,23\}$ of the visual encoder. We train the model end-to-end, fully training the CLFA parameters while fine-tuning all linear layers of the MLLM using LoRA (Hu et al., 2022) (rank = 16, $\alpha = 32$). In total, 96.6M parameters are involved in training. The model is trained for 1 epoch with a learning rate of 2e-4, batch size of 8, $G = 4$, $\varepsilon = 0.2$, $\beta = 0.04$, and temperature 0.9. Experiments use a fixed random seed of 42, and training on two A100 GPUs takes approximately 3 hours. These hyperparameters are set empirically; although we do not conduct hyperparameter ablation, the model trained with this configuration already achieves state-of-the-art performance. Notably, we do not apply any data augmentation to the training data to ensure fair comparison.

## 4.2 Generalization to Unseen Generative Models

We compare our method with 11 state-of-the-art AI-generated image detectors on AIGIBench (Li et al., 2025c). All detectors are trained on the same AIGIBench training set, which contains images generated by SD-v1.4 (Rombach et al., 2022) and ProGAN (Karras et al., 2018). Testing is conducted on 25 subsets covering a variety of generator types, focusing on the detectors' generalization to image sources unseen during training. Table 1 reports the Average Precision (A.P.) for each subset. Our method achieves a mean A.P. of 85.9%, surpassing the previous SOTA method AIDE (Yan et al., 2025b) by 3.2%. For images generated by different diffusion models, our method consistently maintains high performance. Notably, on DALLE-3 generated images, our method achieves 99.6% A.P., significantly outperforming previous methods and demonstrating strong generalization to unseen generation sources. For GAN-based deepfake images, our method shows relatively limited per-

Table 2: Comparison with state-of-the-art MLLM on image modality of LOKI benchmark. We report the accuracy (%) on each subset and the overall accuracy. * denotes the closed-source models. Except for the results of Qwen2.5-VL-7B and our method, all other results are taken from the LOKI benchmark paper (Ye et al., 2025).

| Method | Overall | Scene | Animal | Person | Object | Medicine | Doc | Satellite |
|---|---|---|---|---|---|---|---|---|
| Human | 27.3 | 24.0 | 25.8 | 19.9 | 26.9 | 26.1 | 22.1 | - |
| Expert(AIDE) | 63.1 | - | **89.9** | 62.5 | **96.5** | 53.4 | 49.7 | 39.3 |
| Claude-3.5-Sonnet* | 53.6 | 51.6 | 51.6 | 55.2 | 51.4 | 51.9 | 59.1 | 50.9 |
| Gemini-1.5-Pro* | 43.5 | 53.7 | 35.7 | 51.5 | 30.3 | 50.0 | 47.2 | 38.1 |
| GPT-4o* | 63.4 | 70.1 | 69.7 | **84.4** | 70.3 | 54.3 | 60.1 | 45.0 |
| MiniCPM-V-2.6 | 44.8 | 52.0 | 34.4 | 53.1 | 31.5 | 53.8 | 51.5 | 38.3 |
| Phi-3.5-Vision | 52.5 | 50.8 | 41.7 | 71.5 | 34.1 | 57.3 | 54.3 | 60.5 |
| LLaVA-OneVision-7B | 49.8 | 59.2 | 41.9 | 58.1 | 37.3 | 52.3 | 53.0 | 50.1 |
| InternLM-XComposer2.5 | 46.4 | 52.7 | 40.0 | 56.7 | 32.5 | **56.1** | 49.8 | 38.2 |
| mPLUG-Owl3-7B | 45.9 | 52.1 | 37.3 | 52.9 | 31.4 | 55.3 | 53.8 | 38.1 |
| Qwen2-VL-7B | 47.8 | 54.7 | 38.9 | 57.9 | 30.3 | 56.0 | 59.6 | 36.9 |
| LongVA-7B | 46.2 | 57.6 | 37.4 | 52.5 | 34.1 | 54.4 | 49.8 | 39.7 |
| Mantis-8B | 54.6 | 54.9 | 52.2 | 54.8 | 53.5 | 53.1 | 51.9 | **63.3** |
| Idefics2-8B | 45.0 | 51.8 | 35.3 | 52.3 | 29.2 | 52.3 | 53.9 | 40.6 |
| InternVL2-8B | 49.7 | 58.8 | 39.4 | 54.4 | 37.8 | 53.9 | 60.2 | 44.2 |
| Llama-3-LongVILA-8B | 49.8 | 49.8 | 50.5 | 50.6 | 47.2 | 50.0 | 49.9 | 50.0 |
| VILA1.5-13B | 49.3 | 52.0 | 38.6 | 54.2 | 31.0 | 50.1 | 56.6 | 62.4 |
| InternVL2-26B | 44.3 | 51.6 | 35.4 | 50.8 | 28.2 | 51.3 | 54.4 | 37.6 |
| VILA1.5-40B | 48.8 | 53.7 | 39.3 | 50.0 | 33.4 | 52.5 | 59.9 | 50.6 |
| InternVL2-40B | 49.6 | 55.7 | 37.3 | 59.2 | 34.8 | 55.5 | 64.8 | 40.8 |
| Qwen2-VL-72B | 53.2 | 55.9 | 43.4 | 66.9 | 38.0 | 55.9 | 73.7 | 38.2 |
| LLaVA-OneVision-72b | 46.3 | 54.7 | 31.6 | 53.1 | 27.8 | 52.1 | **67.9** | 36.6 |
| Qwen2.5-VL-7B | 53.8 | 53.7 | 53.2 | 55.6 | 51.9 | 54.3 | 58.5 | 50.9 |
| **Ours** | **69.1** | **71.3** | 84.1 | 66.3 | 85.5 | 53.3 | 52.6 | 62.9 |

formance, which is expected due to the significant differences between such face-swap images and fully generated images in the training set. Importantly, the SocialRF and CommunityAI subsets contain synthetic images from social platforms and art communities, respectively. Our method achieves 98.9% and 87.0% A.P. on these subsets, further validating its effectiveness in real-world scenarios. Additionally, we report accuracy metrics on AIGIBench in Appendix B, where our method achieves mean accuracy comparable to previous SOTA methods.

Table 2 presents the performance comparison of our method with existing multimodal large language models (MLLMs) on the image judgement task of the LOKI Benchmark (Ye et al., 2025). Overall accuracy is calculated as a weighted average, where the weights correspond to the number of images in each subset. Our method uses Qwen2.5-VL-7B as the base model and undergoes reinforcement fine-tuning on the AIGIBench training set, while results for other MLLMs reflect their zero-shot performance. Our method achieves an overall accuracy of 69.1%, surpassing the current state-of-the-art MLLM GPT-4o by 5.7% and improving 15.3% over the base model. On the Scene, Animal, and Object subsets, our method achieves the highest accuracy. This comparison demonstrates that our method effectively elicit the intrinsic potential of MLLMs, enabling more effective AI-generated image detection for unseen generative models.

## 4.3 ROBUSTNESS UNDER VARIOUS IMAGE DEGRADATIONS

Table 3 presents the robustness evaluation results on AIGIBench, including JPEG compression, Gaussian noise, and up/down-sampling. Specifically, we apply JPEG compression with a quality factor of 50 to simulate compression artifacts; add Gaussian noise with standard deviation $\sigma = 4$; and downsample the images to half of their original size using nearest-neighbor interpolation, then upsample back to the original resolution to simulate practical sampling processes. Overall, our method maintains the best detection performance under various image post-processing conditions. Notably, the performance of all AI-generated image detectors drops significantly under JPEG compression. Only the CLIP-based detectors CLIPD (Ojha et al., 2023) and Effort (Yan et al., 2025c)

Table 3: The overall robust performance of AI-generated image detectors in real- world scenarios. We report the average performance, averaged across all subsets of AIGIBench. R.Acc and F.Acc denote real accuracy (on natural images) and fake accuracy (on AI-generated images), respectively.

| Method | Origin | | | | JPEG Compression | | | | Gaussian Noise | | | | Up-down Sampling | | | |
|---|---|---|---|---|---|---|---|---|---|---|---|---|---|---|---|---|
| | R.Acc | F.Acc | Acc | A.P. | R.Acc | F.Acc | Acc | A.P. | R.Acc | F.Acc | Acc | A.P. | R.Acc | F.Acc | Acc | A.P. |
| CNND | 98.2 | 11.6 | 55.1 | 67.0 | 94.3 | 17.2 | 55.8 | 63.7 | 97.7 | 2.6 | 50.2 | 47.0 | 99.8 | 1.8 | 50.8 | 56.7 |
| Gram-net | 90.5 | 26.6 | 58.6 | 62.4 | 99.6 | 1.2 | 50.4 | 55.8 | 95.4 | 10.6 | 53.0 | 60.5 | 91.2 | 25.1 | 58.2 | 63.9 |
| LGrad | 85.8 | 39.6 | 62.9 | 66.6 | 95.9 | 7.3 | 51.6 | 54.6 | 91.9 | 17.5 | 54.7 | 60.0 | 86.5 | 57.2 | 71.9 | 80.3 |
| CLIPD | 73.3 | 71.5 | 72.5 | 75.6 | 91.1 | 33.0 | 62.1 | 71.6 | 78.3 | 58.7 | 68.5 | 72.2 | 77.0 | 66.6 | 71.8 | 75.0 |
| FreqNet | 65.9 | 66.4 | 66.2 | 70.1 | 99.5 | 1.4 | 50.5 | 53.0 | 73.7 | 48.5 | 61.1 | 66.2 | 74.7 | 63.1 | 68.9 | 73.2 |
| NPR | 93.8 | 41.9 | 67.9 | 73.9 | 100.0 | 0.2 | 50.1 | 59.2 | 98.5 | 6.2 | 52.4 | 68.5 | 94.8 | 34.3 | 64.6 | 81.0 |
| DFFreq | 89.6 | 51.9 | 71.1 | 75.7 | 100.0 | 0.1 | 50.1 | 58.8 | 86.3 | 32.2 | 59.3 | 69.0 | 91.8 | 41.9 | 66.9 | 75.3 |
| LaDeDa | 91.7 | 54.9 | 73.4 | 79.3 | 100.0 | 0.0 | 50.0 | 61.6 | 98.8 | 2.6 | 50.7 | 68.5 | 92.2 | 46.6 | 69.4 | **84.5** |
| AIDE | 88.1 | 67.0 | 77.6 | 82.7 | 98.9 | 1.5 | 50.2 | 50.3 | 93.0 | 22.4 | 57.7 | 72.5 | 74.8 | 27.4 | 51.1 | 55.1 |
| SAFE | 96.8 | 63.0 | **79.9** | 82.6 | 100.0 | 0.0 | 50.0 | 48.7 | 100.0 | 1.2 | 50.6 | 46.9 | 100.0 | 16.2 | 58.1 | 73.5 |
| CO-SPY | 71.4 | 73.5 | 72.5 | 79.1 | 94.5 | 40.2 | 67.3 | 80.9 | 85.9 | 55.5 | 70.7 | 77.9 | 62.3 | 76.0 | 69.1 | 75.8 |
| Effort | 84.6 | 65.7 | 75.2 | 81.0 | 98.3 | 31.2 | 64.7 | 75.1 | 91.0 | 49.3 | 70.2 | 80.2 | 75.4 | 72.0 | 73.7 | 80.0 |
| **Ours** | 91.9 | 67.7 | 79.8 | **85.9** | 98.9 | 39.7 | **69.3** | **81.7** | 93.0 | 58.7 | **75.8** | **84.0** | 84.8 | 65.4 | **75.1** | 82.1 |

Table 4: Ablation study of key components. Experiments are conducted on AIGIBench, and we report the mean performance across all subsets. LoRA F.T. denotes LoRA fine-tuning, while an em dash (——) indicates that the module is frozen. All other training settings are kept the same.

| | Method | CLFA | ViT | Projector | LLM | R.Acc. | F.Acc. | Acc. | A.P. |
|---|---|---|---|---|---|---|---|---|---|
| 1 | Zero-Shot | —— | —— | —— | —— | 93.1 | 35.2 | 64.2 | 77.4 |
| 2 | +SFT | —— | —— | LoRA F.T. | LoRA F.T. | 62.5 | 76.4 | 69.5 | 75.8 |
| 3 | +GRPO | —— | —— | LoRA F.T. | LoRA F.T. | 91.2 | 59.2 | 75.2 | 82.1 |
| 4 | +GRPO | —— | LoRA F.T. | LoRA F.T. | LoRA F.T. | 91.5 | 65.4 | 78.5 | 84.7 |
| 5 | +GRPO | ✓ | LoRA F.T. | LoRA F.T. | LoRA F.T. | 91.9 | 67.7 | **79.8** | **85.9** |

retain a certain level of F.Acc. Artifact-based detectors see their detection accuracy on AI-generated images (F.Acc.) fall close to 0%. These detectors rely on unique textures left by generators in images to detect AI content, but JPEG compression substantially weakens these artifacts. Our method utilizes the visual encoder of the MLLM, allowing it to better resist JPEG compression, highlighting its robustness and reliability in real-world scenarios.

### 4.4 ABLATION STUDIES

In this section, we conduct an ablation study on the key components of our method. Table 4 presents the ablation results on AIGIBench. The first row corresponds to the zero-shot performance of the base model Qwen2.5-VL-7B, which shows relatively low accuracy (35.2%) on AI-generated image detection. In the second and third rows, we apply supervised fine-tuning (SFT) and GRPO, respectively, with LoRA applied to the projector and LLM. Both methods only access input images and their ground-truth labels "real/fake" during training. We observe that SFT tends to overfit to images in the training set, whereas GRPO promotes better generalization. In the fourth and fifth rows, we successively incorporate LoRA fine-tuning on the ViT and our designed Cross-Layer Forensic Adapter (CLFA). Parallel integration of CLFA in the visual encoder allows the model to fully leverage multi-level visual features for image authenticity judgment, achieving the best performance with 79.8% Acc. and 85.9% A.P. In Appendix C, we further compare our CLFA with a direct additive fusion of ViT intermediate features. In Appendix D, we conduct ablation experiments on the base model, showing that our method is effective across different model sizes.

### 4.5 CASE STUDIES

We conduct a case study to gain deeper insights into our model's behavior. In Figure 3a, the model successfully captures subtle reasoning cues that indicate an image is AI-generated, allowing it to correctly classify the image as fake. In contrast, Figure 3b shows a failure case, where the model

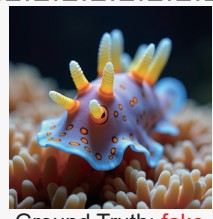

<think> The image depicts a sea slug with vibrant colors and intricate details, which are characteristic of many real sea slugs found in nature. However, the level of detail and the way the light interacts with the slug's surface suggest a high level of realism that could be achieved through advanced digital rendering techniques. The background also appears to be a natural underwater environment, but the clarity and sharpness of the image give it a slightly artificial look.</think>
<answer> **fake** </answer>                                           fake_prob: 0.9922

Ground Truth: fake

(a) Success case: The model detects AI-generated clues and classifies the image as AI-generated.

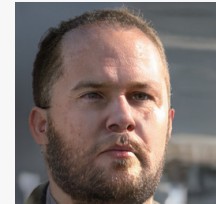

<think> The image appears to be of a real person based on the natural skin texture, facial features, and overall composition. There are no obvious signs of digital manipulation or artificial elements that would suggest it is AI-generated. The lighting and shadows also seem consistent with a real photograph.</think>
<answer> **real** </answer>                                           fake_prob: 0.0025

Ground Truth: fake

(b) Failure case: The model misses AI-generated clues and mistakenly classifies the image as real.

Figure 3: Case studies of our method on success and failure cases.

incorrectly classifies an AI-generated image as real. This error highlights the challenges faced by current AIGI detectors, particularly when synthetic images exhibit highly realistic textures and lacks obvious visual flaws. This analysis clarifies the strengths and limitations of our method and offers directions for developing more robust detectors in real-world scenarios.

## 5 RELATED WORK

### 5.1 AI-GENERATED IMAGE DETECTION

Previous non-MLLM AI-generated image detection methods (Wang et al., 2020; Liu et al., 2020; Tan et al., 2023) mostly train a binary classifier to distinguish between natural and AI-generated images. CLIP-based detectors (Ojha et al., 2023; Yan et al., 2025c) train a linear classifier on image embeddings extracted by a CLIP visual encoder. Artifact-based detectors (Tan et al., 2024b; Yan et al., 2025a; Li et al., 2025a) first extract artifact representations from low-level image signals and then train a binary classifier on these representations. AIDE (Yan et al., 2025b) combine a CLIP branch with an artifact extraction branch to integrate the two types of detectors. Ji et al. (2025) has proposed systematic prompting strategies to enhance the AI-generated image detection capability of MLLMs, while Li et al. (2025b); Wen et al. (2025); Zhou et al. (2025) further improve MLLMs through supervised fine-tuning; however, these approaches require costly and time-consuming textual annotations. Recently, Huang et al. (2025) enhances the AI-generated image detection capability of MLLMs through RL-based post-training, thereby avoiding costly textual annotations.

### 5.2 POST-TRAINING OF MULTIMODAL LARGE LANGUAGE MODELS

In recent years, Multimodal Large Language Models (MLLMs) have demonstrated strong capabilities across various domains, and post-training methods such as supervised fine-tuning (SFT) and reinforcement learning (RL) have rapidly developed. DeepSeekMath (Shao et al., 2024) introduces Group Relative Policy Optimization (GRPO), a variant of Proximal Policy Optimization (PPO) (Schulman et al., 2017), which effectively enhances LLM reasoning abilities.Recent works (Chen et al., 2025; Liu et al., 2025) extend GRPO to MLLMs, demonstrating strong generalization potential on tasks such as image classification and object detection; meanwhile, Xu et al. (2025) enhances the human-centric video forgery detection capability of MLLMs through GRPO-based post-training.

## 6 CONCLUSION

We propose AIGID-RFT, a novel MLLM-based method for AI-generated image detection. By introducing CLFA and GRPO, our method fully utilizes multi-level features from the visual encoder and unlocks the intrinsic AIGI detection potential of MLLMs. Extensive experiments on AIGIBench demonstrate that existing detectors possess some generalization ability but lack robustness, whereas our method maintains optimal performance under various image degradation conditions. Although AI-generated image detection remains far from solved, our method lays the foundation for building more reliable detectors.

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

## A    PROMPT USED IN OUR METHOD

In our method, we use the following prompt during training and testing on AIGIBench. For evaluation on the LOKI benchmark, we strictly follow its official code and prompts.

> **Prompt**
>
> This is an image. Please identify whether the image is real or AI-generated. Output the thinking process in <think> </think> and final answer in <answer> </answer> tags. The output answer format should be as follows:
> <think> ... </think> <answer>real|fake</answer> Please strictly follow the format.

## B    ACCURACY RESULTS ON AIGIBENCH

Table 5: Comparison with state-of-the-art AI-generated image detectors on AIGIbench. The table reports the accuracy (Acc.) for each of the 25 subsets, and the mean accuracy across all subsets. All methods are trained on the same dataset containing images from SD-v1.4 and ProGAN. The best and the second-best performance are indicated by **bold** and underline, respectively.

| Method | CNND | Gram-net | LGrad | CLIPD | FreqNet | NPR | DFFreq | LaDeDa | AIDE | SAFE | CO-SPY | Effort | **Ours** |
| --- | --- | --- | --- | --- | --- | --- | --- | --- | --- | --- | --- | --- | --- |
| Year | 2020 | 2020 | 2023 | 2023 | 2024 | 2024 | 2024 | 2025 | 2025 | 2025 | 2025 | 2025 | 2025 |
| *Diffusion for Text-to-Image Generation* | | | | | | | | | | | | | |
| FLUX1-dev | 57.4 | 64.5 | 80.4 | 80.0 | 78.5 | 95.2 | 76.6 | 94.6 | 88.0 | 98.1 | 77.7 | 75.4 | 93.6 |
| Midjourney-V6 | 52.3 | 43.9 | 60.4 | 65.3 | 53.9 | 68.8 | 64.6 | 80.3 | 76.4 | 94.1 | 69.4 | 73.8 | 88.1 |
| GLIDE | 51.1 | 71.6 | 82.4 | 76.7 | 75.8 | 82.5 | 88.5 | 87.1 | 93.4 | 92.5 | 75.2 | 83.6 | 70.3 |
| DALLE-3 | 53.9 | 53.5 | 57.2 | 75.1 | 66.2 | 57.1 | 51.8 | 50.1 | 55.1 | 49.0 | 81.1 | 77.0 | 95.0 |
| Imagen3 | 51.2 | 50.6 | 62.2 | 78.9 | 73.6 | 85.9 | 75.7 | 91.6 | 89.8 | 96.7 | 79.1 | 65.6 | 96.0 |
| SD3 | 55.8 | 52.5 | 63.4 | 84.5 | 77.3 | 91.9 | 81.3 | 95.1 | 94.3 | 94.1 | 89.3 | 82.4 | 96.5 |
| SDXL | 52.8 | 63.8 | 73.6 | 84.7 | 82.7 | 86.6 | 88.9 | 94.7 | 93.5 | 98.3 | 85.4 | 79.3 | 96.5 |
| BLIP | 77.2 | 98.6 | 93.0 | 88.6 | 93.8 | 99.2 | 97.9 | 99.0 | 96.4 | 99.7 | 85.0 | 98.0 | 99.0 |
| *Diffusion for Personalized Generation* | | | | | | | | | | | | | |
| Infinite-ID | 49.7 | 50.6 | 50.9 | 84.5 | 79.0 | 63.9 | 70.4 | 61.5 | 92.2 | 96.9 | 89.8 | 92.2 | 95.0 |
| InstantID | 53.2 | 75.1 | 72.6 | 85.4 | 79.8 | 63.8 | 91.9 | 86.5 | 91.8 | 98.2 | 84.0 | 92.2 | 93.6 |
| IP-Adapter | 52.0 | 54.5 | 70.3 | 82.6 | 78.8 | 82.4 | 83.2 | 90.8 | 90.0 | 92.8 | 74.8 | 87.5 | 91.6 |
| PhotoMaker | 50.1 | 50.1 | 59.9 | 69.3 | 77.0 | 48.1 | 88.0 | 78.4 | 91.7 | 97.0 | 55.8 | 76.6 | 86.1 |
| *GAN-based Noise-to-Image Generation* | | | | | | | | | | | | | |
| ProGAN | 97.6 | 98.5 | 96.6 | 98.4 | 99.3 | 99.4 | 98.1 | 99.8 | 97.2 | 100.0 | 96.6 | 100.0 | 90.1 |
| R3GAN | 50.4 | 47.9 | 54.4 | 83.5 | 62.3 | 50.8 | 61.7 | 54.8 | 92.9 | 93.9 | 84.5 | 90.2 | 57.9 |
| StyleGAN3 | 55.8 | 65.6 | 70.5 | 79.6 | 83.0 | 78.1 | 90.1 | 92.4 | 88.1 | 89.7 | 84.5 | 87.9 | 92.6 |
| StyleGAN-XL | 52.8 | 72.9 | 65.7 | 84.6 | 79.8 | 60.3 | 73.7 | 94.7 | 88.7 | 93.1 | 85.4 | 92.6 | 68.8 |
| StyleSwim | 52.6 | 75.5 | 81.3 | 86.4 | 80.8 | 85.7 | 84.5 | 94.0 | 83.7 | 97.8 | 79.6 | 94.5 | 88.2 |
| WFIR | 49.8 | 44.5 | 51.7 | 70.0 | 58.5 | 51.6 | 74.6 | 58.5 | 71.4 | 60.4 | 74.8 | 85.5 | 83.7 |
| *GANs for Deepfake* | | | | | | | | | | | | | |
| BlendFace | 52.4 | 42.3 | 41.8 | 35.0 | 23.3 | 44.5 | 41.5 | 42.4 | 51.5 | 47.3 | 41.3 | 41.4 | 44.1 |
| E4S | 51.1 | 42.5 | 41.5 | 57.0 | 25.8 | 45.0 | 42.1 | 42.9 | 44.3 | 47.6 | 68.0 | 49.6 | 59.4 |
| FaceSwap | 50.3 | 47.0 | 45.3 | 53.1 | 40.4 | 48.1 | 47.6 | 47.1 | 52.1 | 50.7 | 50.0 | 48.8 | 50.0 |
| InSwap | 54.5 | 47.1 | 44.6 | 43.7 | 37.5 | 47.8 | 47.0 | 47.0 | 50.9 | 49.7 | 36.4 | 48.0 | 49.5 |
| SimSwap | 52.1 | 46.7 | 44.2 | 43.7 | 36.5 | 47.4 | 45.6 | 46.3 | 54.9 | 49.0 | 39.8 | 52.3 | 49.5 |
| *Open-source Platforms* | | | | | | | | | | | | | |
| SocialRF | 51.1 | 52.1 | 53.5 | 54.4 | 54.2 | 59.1 | 57.6 | 58.6 | 57.8 | 58.0 | 61.7 | 53.1 | 94.6 |
| CommunityAI | 51.3 | 52.6 | 55.5 | 67.0 | 55.9 | 54.0 | 54.5 | 54.5 | 54.1 | 54.2 | 62.6 | 51.6 | 67.8 |
| **Mean Acc.** | 55.1 | 58.6 | 62.9 | 72.5 | 66.2 | 67.9 | 71.1 | 73.4 | 77.6 | **79.9** | 72.5 | 75.2 | 79.8 |

Table 5 reports the accuracy on each subset of AIGIBench. Our method achieves a mean accuracy of 79.8%, which is comparable to the previous method SAFE (Li et al., 2025a). Note that this performance is on the original images; under various post-processing operations, our method clearly surpasses previous methods.

## C    COMPARISON BETWEEN CLFA AND DIRECT ADDITIVE FUSION

We further conduct an ablation study on CLFA by comparing it with direct addition fusion, as shown in Table 6. Specifically, "direct addition fusion" directly adds the intermediate features to the final

image features:

$$\mathbf{F}_{\text{img}} = \mathbf{F}_{\text{img}}^4 + \sum_{i=1}^{3} \mathbf{F}_{\text{img}}^i. \tag{9}$$

In contrast, our proposed CLFA adapts the intermediate features before integration, thereby achieving better detection performance.

Table 6: Comparison between CLFA and direct additive fusion on AIGIBench

| Method | R.Acc. | F.Acc. | Acc. | A.P. |
|---|---|---|---|---|
| direct addition fusion | 91.3 | 65.1 | 78.2 | 84.8 |
| CLFA (Ours) | 91.9 | 67.7 | 79.8 | 85.9 |

## D  ABLATION STUDY OF DIFFERENT MODEL SIZES

Ablation experiments on different base models are presented in Table 7, showing that our method is effective across models of various sizes.

Table 7: Ablation study of different model sizes on AIGIBench

| Method | R.Acc. | F.Acc. | Acc. | A.P. |
|---|---|---|---|---|
| Qwen2.5-VL-3B (zero-shot) | 78.5 | 44.6 | 61.5 | 67.3 |
| Qwen2.5-VL-3B+Ours | 82.5 | 69.1 | 75.8 | 83.9 |
|  |  |  | +14.3 | +16.6 |
| Qwen2.5-VL-7B (zero-shot) | 93.1 | 35.2 | 64.2 | 77.4 |
| Qwen2.5-VL-7B+Ours | 91.9 | 67.7 | 79.8 | 85.9 |
|  |  |  | +15.6 | +8.5 |

## E  COMPARISON BETWEEN THINK AND NO-THINK

In Table 8, we conduct an ablation study on the inference process. Using the prompt below, we instruct the model to output only the answer without generating the content between the `<think>` tag. This experiment demonstrates that including the reasoning process can improve the accuracy of AI-generated image detection.

---
**Prompt no think**

This is an image. Determine if it is real or AI-generated.
Output the final decision only inside `<answer>...  </answer>` tags.
The valid outputs are exactly one of the following:
`<answer>real</answer> <answer>fake</answer>`
Do not output any reasoning or explanations.

---

Table 8: Comparison between think and no-think

| Method | R.Acc. | F.Acc. | Acc. | A.P. |
|---|---|---|---|---|
| no-think | 94.8 | 62.8 | 78.8 | 85.3 |
| think (Ours) | 91.9 | 67.7 | 79.8 | 85.9 |

## F    COMPARISON OF INFERENCE SPEED

We compare the inference speeds of our method with recent methods, CO-SPY Cheng et al. (2025) and Effort (Yan et al., 2025c), with the results shown below. The experiments are conducted on a single A100 GPU with a batch size of 1. The relatively lower inference speed of our method is expected, as it is based on an MLLM, which generates both the reasoning process and the final answer.

Table 9: Inference speed comparison of different methods on AIGIBench

| Model | FPS |
| --- | --- |
| CO-SPY (CVPR2025) | 14.51 |
| Effort (ICML2025) | 15.09 |
| Ours (based on Qwen2.5-VL-7B) | 0.284 |

## G    ETHICS STATEMENT

Ethics Statement This work uses only publicly available datasets and does not involve human subjects or sensitive personal data. The goal is to improve AI-generated image detection for digital forensics and trustworthy media, thereby mitigating risks of misinformation and malicious synthetic content.

## H    REPRODUCIBILITY STATEMENT

We fix the random seed to 42 in all experiments, and all datasets used in this work are publicly available. Implementation details and hyperparameters are provided in Section 4.1. If the paper is accepted, we will release the complete code and models to further support reproducibility.

## I    LIMITATION

Due to computational constraints, we experiment with models up to 7B parameters and apply LoRA fine-tuning to reduce memory usage. Additionally, since our method is based on MLLMs, inference is slower compared to previous non-MLLM methods. This work mainly focuses on detection performance and robustness, while a deeper analysis of the model's reasoning process is left for future work.

## J    LLM USAGE STATEMENT

In this paper, we use LLMs solely to improve grammar and wording.

