# OpenReview forum: "AIGID-RFT: Reinforcement Fine-Tuning Multimodal LLMs for AI-Generated Image Detection"
_ICLR.cc/2026/Conference — Submitted to ICLR 2026_

### Official Review · Reviewer_pa3X · 2025-10-25

**Soundness:** 3
**Presentation:** 3
**Contribution:** 3
**Rating:** 8
**Confidence:** 4

**Summary:**

This paper proposes AIGID-RFT, a novel MLLM-based AI-generated image (AIGI) detector that leverages reinforcement learning with verifiable rewards (GRPO) and a Cross Layer Forensic Adapter (CLFA). The method enhances robustness to real-world post-processing operations - particularly JPEG compression (QF=50) - by integrating CLFA in parallel across intermediate layers of the vision encoder. Experiments on AIGIBench and LOKI show that AIGID-RFT achieves state-of-the-art performance, especially under degradation, while requiring only binary "real / fake" labels for training.

[This review was utilising LLMs for typos and grammar/wording improvements, as well as literature research.]

**Strengths:**

1) Reproducible Setup: The use of public datasets (AIGIBench), open models (Qwen2.5-VL-Instruct 3B and 7B), fixed random seed (42), and full implementation details.
2) Novel methodological integration: Combining reinforcement learning with MLLMs for AIGI detection is underexplored; this work fills an important gap.
3) Strong evaluation design: Includes ablations, robustness tests across degradations, generalization to unseen generators (e.g., DALLE-3), and comparison to 11 SOTA detection methods.
4) Clear motivation and alignment with real-world needs (JPEG compression), which remains a major challenge in practice.
5) Regulatory and societal relevance: With regulations like the EU AI Act mandating transparency for AI-generated content, robust detection methods are becoming legally necessary. This work addresses a critical infrastructure need as synthetic media becomes ubiquitous across journalism, legal proceedings, and public discourse.

**Weaknesses:**

While the paper is technically strong, it falls short in contextualising its contribution relative to recent peer-reviewed SOTA methods (published before July 24, 2025, see last Q/A in https://iclr.cc/Conferences/2026/ReviewerGuide):
1) Missing comparison with CO-SPY (CVPR 2025): a recently released, highly relevant work that also targets generalization and robustness in AIGI detection by combining semantic and artifact-based features. The ICLR policy requires authors to cite and compare with any peer-reviewed paper published before July 24, 2025. This comparison is essential for evaluating the novelty and impact of AIGID-RFT.
2) No discussion or comparison with existing forensic adapters, such as Cui et al., "Forensics Adapter: Adapting CLIP for Generalizable Face Forgery Detection" (CVPR 2025), which introduces adapter-based fine-tuning in visual encoders.
3) While comparisons of CLFA vs Direct Additive Fusion (Appendix C), different model sizes (Appendix D) and prompts (think vs no-think, Appendix E) were studied, a direct ablation of the vision encoder + CLFA alone vs. the full MLLM pipeline would help clarify whether the performance gain comes from the adapter or is due to broader model capabilities.
4) While inference speed and scalability are acknowledged briefly, a more nuanced analysis (e.g., latency comparison with lightweight CNNs, or an accuracy/compute metric for all tested models) would strengthen the practical impact statement.
5) The comparison to the base MLLMs is done using zero-shot performance. A comparison to the models capabilities using few-shot prompts (see also theoretical discussions in https://arxiv.org/html/2507.16003v1), would also raise the quality of your new methodology.

**Questions:**

To help clarify the contribution and improve the paper’s scientific rigor and actuality, please address:
1) How does AIGID-RFT compare quantitatively to CO-SPY (CVPR 2025)
2) Could you include an ablation that evaluates only the vision encoder + CLFA (without the LLM)? This would help isolate whether the gains come from the adapter itself or depend on full MLLM integration.

---

> ### Author Response · Authors · 2025-11-24
>
> Dear reviewer pa3X, we sincerely appreciate your thoughtful comments, and your suggestions have been very helpful in improving the quality of our manuscript. Please find our point-by-point responses below.
>
>
>
> ## **Questions:**
>
>
>
> **Q1**: How does AIGID-RFT compare quantitatively to CO-SPY (CVPR 2025)
>
> **A1**: We have added a quantitative comparison with CO-SPY [1] and incorporated the results into the revised manuscript (Table 3).  The implementation of CO-SPY strictly follows the authors’ released code, and all three of its training stages are trained for one epoch on the AIGIBench training set. Overall, our model achieves superior performance compared to CO-SPY.
>
>
>
> ###### **Table D (corresponding to Table 3 in the manuscript): Comparison results (ACC/AP) with CO-SPY on AIGIBench.**
>
> | Model  | Origin    | JPEG Compression |
> | ------ | --------- | ---------------- |
> | CO-SPY | 72.5/79.1 | 67.3/80.9        |
> | Ours   | 79.8/85.9 | 69.3/81.7        |
>
>
>
> In addition, we have included a comparison of inference time. The experiments are conducted on a single A100 GPU with a batch size of 1. The relatively lower inference speed of our method is expected, as it is based on a MLLM, which generates both the reasoning process and the final answer.
>
> ###### **Table A: Inference speed comparison of different methods on AIGIBench**
>
> | Model                         | FPS (frames per second) |
> | ----------------------------- | ----------------------- |
> | CO-SPY (CVPR2025)             | 14.51                   |
> | Effort (ICML2025)             | 15.09                   |
> | Ours (based on Qwen2.5-VL-7B) | 0.284                   |
>
>
>
> For [2], its training requires masks of the tampered regions for training, and therefore cannot be directly applied or retrained on AIGIBench.
>
>
>
> **Q2**: Could you include an ablation that evaluates only the vision encoder + CLFA (without the LLM)? This would help isolate whether the gains come from the adapter itself or depend on full MLLM integration.
>
> **A2**: Thank you for the suggestion. Removing the LLM would make it impossible to train with GRPO. The visual tokens extracted by a standalone vision encoder have variable lengths that change with image resolution, and the encoder does not provide an explicit class token for classification. This makes our method difficult to implement without the LLM. Our CLFA is specifically designed for the end-to-end GRPO training paradigm, and we believe that the current ablation studies sufficiently demonstrate the performance gains brought by CLFA.
>
>
>
> ------
>
> [1] Cheng et al, CO-SPY: Combining Semantic and Pixel Features to Detect Synthetic Images by AI. CVPR2025
>
> [2] Cui et al, Forensics Adapter: Adapting CLIP for Generalizable Face Forgery Detection. CVPR2025

---

> > ### Comment · Reviewer_pa3X · 2025-11-27
> >
> > Thank you for the detailed responses. The CO-SPY comparison demonstrates clear performance improvements, and your explanation regarding the architectural dependency between CLFA and the MLLM framework is reasonable.
> > Your responses have adequately addressed my concerns.

---

> > > ### Author Response · Authors · 2025-11-28
> > > **Sincere Thanks**
> > >
> > > Thank you for your thoughtful feedback and active engagement throughout the rebuttal process. We sincerely appreciate your support and endorsement!

---

### Official Review · Reviewer_tuqj · 2025-10-27

**Soundness:** 1
**Presentation:** 2
**Contribution:** 1
**Rating:** 2
**Confidence:** 5

**Summary:**

This paper proposes AIGID-RFT, an AIGC Detection method that integrates reinforcement learning and MLLM features. On one hand, it aims to avoid overfitting caused by supervised learning through reinforcement learning. On the other hand, it leverages pre-trained knowledge from MLLMs to fuse features and form more generalizable forensic features. Experiments on AIGIBench demonstrate its effectiveness compared to several existing algorithms.

**Strengths:**

- Proposes an AIGC Detection model that combines reinforcement learning and MLLM pre-trained features.
- Demonstrates effectiveness against SOTA (State-of-the-Art) methods on the AIGIBench and LOKI datasets.

**Weaknesses:**

**Unclear motivation explanation:**

First, why is reinforcement learning adopted, and what are its core advantages over the supervised learning used in existing works? The phenomenon described in lines 038–047 and Figure 1 shows that existing works suffer severe performance degradation when detecting JPEG-compressed data. However, what is the connection between this phenomenon and the use of reinforcement learning? Can reinforcement learning learn to resist interference from JPEG or other post-processing during training? Additionally, does the use of reinforcement learning heavily depend on the initially provided network architecture and weights (e.g., why Qwen2.5-VL-7B is selected)? And why are MLLMs used? It is suggested that the authors first provide a more sufficient explanation of the solution's motivation.

**Weak originality in the method:**

(1) The Cross Layer Forensic Adapter (CLFA) proposed in Section 3.2 is merely an MLP-based feature fusion module. The Up/Down-projection and LayerNorm techniques involved are not original. Furthermore, what is the connection between this adapter and "forensics"? It does not include sufficient improvements tailored to the AIGC Detection task, making it difficult to be regarded as an independent core innovation.

(2) The reinforcement learning-based training stage in Section 3.3 directly adopts GRPO without additional modifications.

**Insufficient experimental analysis:**

(1) The results in Table 1 show poor detection performance for Deepfake (face-swap) images. The paper attributes this to "significant differences between such face-swap images and fully generated images in the training set." If face-swap data is center-cropped, can a significant performance improvement be achieved? It is suggested that the paper conduct a more in-depth discussion on this point.

(2) Lines 356–364 only describe the results in Table 2 without analysis. For example, why does GPT-4o achieve significantly better performance than other MLLMs? Additionally, these MLLMs initially only achieve an accuracy close to random guessing (~0.5), so why is it claimed that MLLMs are beneficial for the AIGC Detection task?

(3) Based on the results in Table 3, why does the proposed method experience a greater performance drop in Fake Accuracy compared to Real Accuracy under JPEG compression? And why can compressed Real images be detected more accurately?

(4) Why do failure cases like those in Figure 3 occur? Is it due to incorrect semantic judgment by the MLLM, or the inability to extract valid forensic traces from visual signals?

(5) Line 416 mentions, "We observe that SFT tends to overfit to images in the training set, whereas GRPO promotes better generalization." Why can GRPO avoid overfitting caused by supervised learning? If other methods (e.g., adversarial training) are adopted, can they achieve similar detection performance to GRPO-based reinforcement learning?

**Questions:**

See Weaknesses.

---

> ### Author Response · Authors · 2025-11-24
>
> Dear reviewer tuqj, we sincerely appreciate your thoughtful comments, and your suggestions have been very helpful in improving the quality of our manuscript. Please find our point-by-point responses below.
>
>
>
>
>
> ## **Unclear motivation explanation:**
>
> **Q1**: Why are MLLMs used? (e.g., why Qwen2.5-VL-7B is selected)?
>
> **A1**: Figure 1 shows that artifact-based detectors exhibit poor robustness, whereas VLM-based detectors (e.g., CLIP-ViT) demonstrate a certain degree of robustness. Based on this observation, our method addresses the robustness issue by adopting Qwen2.5-VL-7B as the base MLLM, whose semantic-level representations offer stronger resistance to post-processing. (We selected Qwen2.5-VL-7B due to its widespread use and its 7B model size, which provides a good balance between performance and computational efficiency.)
>
>
>
> **Q2**: Why is reinforcement learning adopted, and what are its core advantages over the supervised learning used in existing works?
>
> **A2**: Reinforcement learning offers two key advantages over supervised fine-tuning (SFT):
>  (1) it does not require costly textual annotations, as the reward can be computed automatically from model predictions;
>  (2) it optimizes a reward function rather than directly imitating training labels, enabling the model to explore a broader solution space and achieve stronger detection performance.
>
>
>
> **Q3**: Can reinforcement learning learn to resist interference from JPEG or other post-processing during training?
>
> **A3**: Reinforcement learning enhances the model’s detection capability by optimizing behavior based on reward signals. During training, the model learns decision patterns that generalize beyond the training data distribution, which can help it maintain reasonable performance under post-processing operations such as JPEG compression.
>
>
>
> ## **Weak originality in the method:**
>
>
>
> **A1**: The novelty of CLFA lies in its architectural design rather than its MLP components. Our proposed CLFA is parallelly integrated across different intermediate layers of the visual encoder, enabling the LLM to more effectively leverage multi-level visual features, and ablation studies demonstrate its effectiveness. CLFA, together with the base model, is trained end-to-end to develop a expert model for AI-generated image detection.
>
> **A2**: Our method develops a detector with strong generalization and robustness. We emphasize that the main novelty of our work lies in the overall AI-generated image detection methodology, specifically the integration of CLFA with MLLMs and RL based post-training, rather than in the specific RL algorithm (GRPO) itself.

---

> ### Author Response · Authors · 2025-11-24
>
> ## **Insufficient experimental analysis:**
>
> **Q1**: If face-swap data is center-cropped, can a significant performance improvement be achieved?
>
> **A1**:
>
> ###### **Table B: Results on original and center-cropped images (ACC %/AP %)**
>
> |                                       | BlendFace | E4S       | FaceSwap  | InSwap    | SimSwap   |
> | ------------------------------------- | --------- | --------- | --------- | --------- | --------- |
> | **Original images**                   | 44.1/44.4 | 59.4/64.1 | 50.0/58.2 | 49.5/50.0 | 49.5/57.9 |
> | **Center-cropped images (0.5 * 0.5)** | 28.9/39.1 | 37.3/42.4 | 48.5/46.3 | 45.8/53.4 | 44.8/46.3 |
>
> We supplemented the results with center-cropped images, where both the height and width of the original image are cropped to 50%. Based on the results in Table B, we observe a performance drop, which may be attributed to the removal of informative visual content during the cropping process.
>
> In addition, most of the training set consists of objects (e.g., car, cat, chair, horse), with relatively few human face images, and those faces are entirely synthetic. This difference in image distribution also contributes to the suboptimal detection performance.
>
>
>
>
>
> **Q2**: Why does GPT-4o achieve significantly better performance than other MLLMs?  Why is it claimed that MLLMs are beneficial for the AIGC Detection task?
>
> **A2**:
>
>
>
> ###### **Table C: (corresponding to Table 2 in the manuscript) Comparison on LOKI benchmark.**
>
> | Model         | Overall Accuracy (%) |
> | ------------- | -------------------- |
> | Human         | 27.3                 |
> | Expert(AIDE)  | 63.1                 |
> | GPT-4o        | 63.4                 |
> | Qwen2.5-VL-7B | 53.8                 |
> | **Ours**      | **69.1**             |
>
> In Table C (corresponding to Table 2 in the manuscript), except for the results of Qwen2.5-VL-7B and our method, all other results are taken from the LOKI benchmark paper [1]. The LOKI benchmark is highly challenging, with most MLLMs achieving near 0.5 in zero-shot performance. MLLMs are beneficial for the AIGC detection task because (1) they provide the opportunity for language-based explanations; (2) MLLM are scalable, with post-training performance surpassing that of previous expert models.
>
> Since GPT-4o is a closed-source model, its exact training data and scale are unknown. However, its stronger zero-shot performance is likely attributed to more extensive training data and larger model capacity, as commonly observed in foundation models.
>
>
>
>
>
> **Q3**: Why does the proposed method experience a greater performance drop in Fake Accuracy compared to Real Accuracy under JPEG compression?
>
> **A3**: JPEG compression tends to make detectors more likely to classify images as "real," this phenomenon commonly observed in most AI-generated image detectors. As a result, the Fake Accuracy drops sharply, while the Real Accuracy shows a slight increase. Overall, the total detection accuracy still decreases.
>
>
>
> **Q4**: Why do failure cases like those in Figure 3 occur? Is it due to incorrect semantic judgment by the MLLM, or the inability to extract valid forensic traces from visual signals?
>
> **A4**: We believe that MLLMs fail to extract effective forensic traces from visual signals for the following reasons: (1) Some AI-generated images are highly realistic, often without obvious semantic inconsistencies;  (2) there is a distribution shift between the training and testing data. Specifically, since we use only images generated by ProGAN and SDv1.4 in AIGIBench for training, the detector may fail on unseen sources due to the large distribution differences.
>
>
>
> **Q5**:  Why can GRPO avoid overfitting caused by supervised learning? If other methods (e.g., adversarial training) are adopted, can they achieve similar detection performance to GRPO-based reinforcement learning?
>
> **A5**: SFT methods align images in the training set with “real” or “fake” labels and easily overfit the training distribution. In contrast, GRPO optimizes a reward function rather than directly imitating labeled samples, allowing the model to explore diverse decision behaviors and avoid overfitting.
>
> We acknowledge that using adversarial training (or other data augmentation techniques) could potentially improve model performance. To ensure a fair comparison, our experiments strictly follow the AIGIBench setup, using only the original images and corresponding labels from the training set. Under this unified setting, our method achieves the best performance on AIGIBench.
>
>
>
> ------
>
> [1] Ye et al, LOKI: A Comprehensive Synthetic Data Detection Benchmark using Large Multimodal Models. ICLR 2025.

---

### Official Review · Reviewer_hXs7 · 2025-10-31

**Soundness:** 2
**Presentation:** 2
**Contribution:** 2
**Rating:** 2
**Confidence:** 5

**Summary:**

This paper aims to address the robustness of current AI-generated image detectors against post-processing operations by using multimodal large language models. A MLLM-based AI-generated image detector termed AIGID-RFT is proposed by introducing reinforcement learning. Furthermore, a Cross Layer Forensic Adapter is designed with the vision encoder to exploit multi-level visual features for enhanced detection performance. Experiments demonstrate the effectiveness of proposed method.

**Strengths:**

1. This paper leverages the MLLM for AI-generated image detection task, which is interesting.
2. The proposed reinforcement learning with MLLM for the task is interesting.
3. The robustness is actually a critical issue in this area.
4. The visualization example in Figure 3 is practical since it provides some explainable reasoning process.
5. The authors conduct extensive experiments on mainstream MLLM and other AI-generated image detector baselines.

**Weaknesses:**

1. Although the robustness issue the authors aim to solve is critical in this area and the results in Figure. 1 supports this, I am still confused how the authors solve this issue in this paper, more explanations on this are needed.
2. There are already some similar Adapter or LoRA technologies, the authors should explain the difference between their CLFA and these existing ones.
3. There are some more powerful multimodal foundation models being proposed, such as Qwen3-VL, did the authors consider to extend their methods to other powerful recent foundation models?
4. Why choose GRPO for reinforcement learning instread of DPO/PPO? What is the connection between GRPO and CLFA?
5. The authors claim that using reinforcement learning with MLLM for ai-generated detection is unexplored, which I can hardly agree. There are some related work such as [1,2,3]. More discussions and comparions are needed.

[1] Huang, Tai-Ming, et al. "ThinkFake: Reasoning in Multimodal Large Language Models for AI-Generated Image Detection." arXiv preprint arXiv:2509.19841 (2025).
[2] Ji, Yikun, et al. "Towards Explainable Fake Image Detection with Multi-Modal Large Language Models." arXiv preprint arXiv:2504.14245 (2025).
[3] Xu, Zhipei, et al. "AvatarShield: Visual Reinforcement Learning for Human-Centric Video Forgery Detection." arXiv preprint arXiv:2505.15173 (2025).

**Questions:**

Please refer to the weakness part. My major concerns lie in the novelty and method design parts. So I currently lean towards negative ratings, if the authors could address the former issues properly during rebuttal, I will change my ratings.

---

> ### Author Response · Authors · 2025-11-24
>
> Dear reviewer hXs7, we sincerely appreciate your thoughtful comments, and your suggestions have been very helpful in improving the quality of our manuscript. Please find our point-by-point responses below.
>
>
>
> **Q1**: Although the robustness issue the authors aim to solve is critical in this area and the results in Figure. 1 supports this, I am still confused how the authors solve this issue in this paper, more explanations on this are needed.
>
> **A1**: Figure 1 shows that artifact-based detectors exhibit poor robustness, whereas VLM-based detectors (e.g., CLIP-ViT) demonstrate a certain degree of robustness. Based on this observation, our method addresses the robustness issue by adopting Qwen2.5-VL as the base MLLM, whose semantic-level representations offer stronger resistance to post-processing. We further design the CLFA and apply RL-based post-training to transform it into an expert model for AI-generated image detection, thereby enhancing its generalization to previously unseen sources.
>
>
>
>
>
> **Q2**: There are already some similar Adapter or LoRA technologies, the authors should explain the difference between their CLFA and these existing ones.
>
> **A2**: Existing adapter or LoRA technologies mainly focus on fine-tuning the parameters of the original model. In contrast, our proposed CLFA is integrated in parallel within the vision encoder, incorporating intermediate-layer visual features. This feature fusion strengthens the model’s representation ability and leads to improved performance in AI-generated image detection.
>
>
>
> **Q3**: There are some more powerful multimodal foundation models being proposed, such as Qwen3-VL, did the authors consider to extend their methods to other powerful recent foundation models?
>
> **A3**: We acknowledge that more recent MLLMs, such as Qwen3-VL (released on September 24), could potentially achieve even better performance. However, we believe this does not affect the main claim of our paper. Our method is based on Qwen2.5-VL 7B as the base model and already achieves state-of-the-art overall performance on AIGIBench.
>
>
>
> **Q4**: Why choose GRPO for reinforcement learning instread of DPO/PPO? What is the connection between GRPO and CLFA?
>
> **A4**: (1) In the AI-generated image detection task, DPO requires preference data (i.e., paired examples indicating which output is preferred), whereas GRPO and PPO do not. GRPO is a variant of PPO that simplifies the training process by eliminating the need for a value model and instead computes rewards based on a rule function. Considering computational efficiency and training stability, we choose GRPO for our method.
>
> (2) Our proposed CLFA is specifically designed for GRPO. It is integrated in parallel within the vision encoder without disrupting the original model structure. Therefore, no additional pretraining is required, and GRPO training can be performed directly in an end-to-end manner.
>
>
>
> **Q5**: The authors claim that using reinforcement learning with MLLM for ai-generated detection is unexplored, which I can hardly agree. There are some related work such as [1,2,3]. More discussions and comparions are needed.
>
> **A5**: We thank the reviewer for the suggestion to include discussions on related works. Our method differs from [2] in the approach used to enhance the AI-generated image detection capability of MLLMs.  [2] designs various prompts, whereas our method uses RL–based post-training. [3] focuses on human-centric video forgery detection, which differs from the task addressed in our paper, as we aim to detect fully AI-generated images. As for [1], it is submitted to arXiv on September 24, 2025, coinciding with the ICLR submission deadline; we have not seen this paper prior to our submission. Nevertheless, we will include discussions of these three related works in the revised version.
>
>
>
> ------
>
> [1] Huang, Tai-Ming, et al. "ThinkFake: Reasoning in Multimodal Large Language Models for AI-Generated Image Detection." arXiv preprint arXiv:2509.19841 (2025).
>
> [2] Ji, Yikun, et al. "Towards Explainable Fake Image Detection with Multi-Modal Large Language Models." arXiv preprint arXiv:2504.14245 (2025).
>
> [3] Xu, Zhipei, et al. "AvatarShield: Visual Reinforcement Learning for Human-Centric Video Forgery Detection." arXiv preprint arXiv:2505.15173 (2025).

---

### Official Review · Reviewer_dDMg · 2025-11-01

**Soundness:** 4
**Presentation:** 3
**Contribution:** 3
**Rating:** 6
**Confidence:** 3

**Summary:**

This paper proposes AIGID-RFT, a novel approach for detecting AI-generated images using multimodal large language models (MLLMs) fine-tuned with reinforcement learning. The authors note that while current AI-generated image detectors work well on unmodified images, they are fragile against common post-processing(e.g., resizing, compression, filters..). On the other hand, multimodal LLMs have shown impressive general capabilities, but naively applying them to image forensics does not work well. AIGID-RFT addresses this by using reinforcement learning as a post-training multimodal model rather than standard supervised fine-tuning. They develop verifiable reward signals to guide the model toward correct real/fake predictions, and introduce a Cross Layer Forensic Adapter module that is inserted into multiple layers of the model’s visual encoder to capture multi-scale visual features crucial for detecting artifacts. Through an RL training process, the MLLM learns to better recognize AI-generated images and avoid being misled by post-processing tricks. This approach demonstrates how combining the strengths of large multimodal models with RL fine-tuning can improve the deepfake image detection.

**Strengths:**

- I am not expert in this field, but to the best of my knowledge, it is first paper to employ RL post-training for ai generated image detection. Good significance.
- This paper is well written,organized and easy to follow
- Shows strong performance on extensive amounts of experiments. Components are well explained.

**Weaknesses:**

- Lack of inference time comparison with other methods
- Limited literature survey. Since adoption of multimodal model for AIGID is relatively new, it would be better to introduce similar works like https://arxiv.org/pdf/2504.14245. I am not expert in this field but I've seen some similar works before.

**Questions:**

- Is text annotation necessary for SFT method? What if we make label for SFT with simple text like <answer> fake <answer>?
- I think choice of benchmark(AIGIbench, Li et al 2025) is little bit concerning since it is less known benchmark. What is your reasoning behind not using traditional benchmark

---

> ### Author Response · Authors · 2025-11-24
>
> Dear reviewer dDMg, we sincerely appreciate your thoughtful comments, and your suggestions have been very helpful in improving the quality of our manuscript. Please find our point-by-point responses below.
>
>
>
> ## **Questions:**
>
> **Q1**: Is text annotation necessary for SFT method? What if we make label for SFT with simple text like ”\<answer\> fake \<answer\>”?
>
> **A1**: As you mentioned, in our ablation study, the SFT method uses only “\<answer\> real \<answer\>” or “\<answer\> fake \<answer\>” as answers, without any additional textual annotations. Since the training data contains only the final answers, the SFT-tuned model learns to output solely “real” or “fake,” without generating any reasoning process.
>
> **Q2**: I think choice of benchmark(AIGIbench, Li et al 2025) is little bit concerning since it is less known benchmark. What is your reasoning behind not using traditional benchmark.
>
> **A2**: AIGIBench [1] is the latest benchmark for AI-generated image detection, evaluating multiple state-of-the-art detectors on images from 25 different generative sources. We choose this benchmark for two reasons: (1) it provides robustness tests under various post-processing operations, better simulating real-world scenarios; (2) traditional benchmarks have become saturated, whereas on AIGIBench, all methods achieve accuracies below 80%, making it a more challenging and discriminative benchmark for evaluating our AI-generated image detectors.
>
>
>
> ## **Weaknesses:**
>
> **Q3**: Lack of inference time comparison with other methods
>
> **A3**: We compare the inference speeds of our method with recent methods, CO-SPY and Effort, with the results shown below. The experiments are conducted on a single A100 GPU with a batch size of 1. The relatively lower inference speed of our method is expected, as it is based on a MLLM, which generates both the reasoning process and the final answer.
>
>
>
> ###### **Table A: Inference speed comparison of different methods on AIGIBench**
>
> | Model                         | FPS (frames per second) |
> | ----------------------------- | ----------------------- |
> | CO-SPY (CVPR2025)             | 14.51                   |
> | Effort (ICML2025)             | 15.09                   |
> | Ours (based on Qwen2.5-VL-7B) | 0.284                   |
>
>
>
> **Q4**: Limited literature survey. Since adoption of multimodal model for AIGID is relatively new, it would be better to introduce similar works like https://arxiv.org/pdf/2504.14245 [2]. I am not expert in this field but I've seen some similar works before.
>
> **A4**: We thank the reviewer for the suggestion to include discussions on  this related work. We will include a more comprehensive discussion of [2] in the revised manuscript. Our method differs from [2] in the approach used to enhance the AI-generated image detection capability of MLLMs. [2] designs various prompts, whereas our method uses RL–based post-training.
>
>
>
> ------
>
> [1] Li et al, Is Artificial Intelligence Generated Image Detection a Solved Problem? NeurIPS 2025.
>
> [2] Ji et al, Towards Explainable Fake Image Detection with Multi-Modal Large Language Models. ACM MM 2025.

---

### Author Response · Authors · 2025-11-24

We sincerely thank the reviewers once again for their valuable feedback. We have uploaded the revised version of our paper. The main revisions are as follows:

**1**.We have added a comparison with CO-SPY[4].

**2**.We have included inference time comparisons in the supplementary material.

**3**.We have expanded the discussion of related work[1,2,3].





[1] Huang, et al. "ThinkFake: Reasoning in Multimodal Large Language Models for AI-Generated Image Detection." arXiv preprint arXiv:2509.19841 (2025).

[2] Ji, et al. "Towards Explainable Fake Image Detection with Multi-Modal Large Language Models." arXiv preprint arXiv:2504.14245 (2025).

[3] Xu, et al. "AvatarShield: Visual Reinforcement Learning for Human-Centric Video Forgery Detection." arXiv preprint arXiv:2505.15173 (2025).

[4] Cheng, et al, CO-SPY: Combining Semantic and Pixel Features to Detect Synthetic Images by AI. CVPR2025

---

### Comment · Area_Chair_y47U · 2025-11-25

Dear Reviewers,

The authors have submitted their responses to your questions and feedbacks. Please read them and give your comments.

Regards, AC

---

### Author Response · Authors · 2025-12-04
**(For AC) Rebuttal Summary**

Dear Area Chair,

We sincerely appreciate all reviewers for their feedback and constructive suggestions. Below is a brief summary of the clarifications provided in our response and rebuttal.

---

### **Key Strengths Highlighted by the Reviewers**

- **A novel methodology integrating reinforcement learning with MLLMs for AIGI detection**
  *(Reviewers dDMg, hXs7, tuqj, pa3X)*

- **A strong focus on the important problem of robustness**
  *(Reviewers hXs7, pa3X*)

- **Extensive experiments demonstrating the effectiveness of the proposed method**
  *(Reviewers dDMg, hXs7, tuqj, pa3X)*



### **Reviewers’ Main Concerns and Our Responses**

- **Limited literature survey**
  *(Reviewers dDMg, hXs7, pa3X*)
  We have added discussions of the relevant works mentioned by the reviewers.

- **Lack of inference time reporting**
  *(Reviewers dDMg, pa3X)*
  We have added a comparison of inference time.

- **How CLFA differs from existing techniques?**
  *(Reviewers hXs7, tuqj)*
  The novelty of CLFA lies in its parallel integration across multiple intermediate layers of the vision encoder, enabling the LLM to more effectively leverage multi-level visual representations.

- **Why choose GRPO?**
  *(Reviewers hXs7, tuqj)*
  We have clarified the motivation for choosing GRPO, primarily because it avoids the need for additional text annotations.

- **Insufficient experimental analysis**
  *(Reviewer tuqj)*
  We have added the requested additional experiments along with the corresponding analyses.

- **Comparison with CO-SPY (CVPR 2025)**
  *(Reviewer pa3X)*
  We have added a comparison with CO-SPY, and the results demonstrate the advantages of our approach.



### **Conclusion**

We believe that the reviewers’ concerns have been thoroughly addressed. Reviewer *pa3X* explicitly stated that our responses have adequately addressed the concerns. Reviewer *hXs7* noted in the original review that they would be willing to change the rating if the concerns were properly addressed, and we hope that our clarifications meet this expectation. Unfortunately, due to the abrupt end of the rebuttal period, we did not receive further feedback from reviewers *dDMg*, *hXs7*, and *tuqj*.

Our work aims to address an increasingly important problem in AI safety: ensuring the robustness of AI-generated image detection. We respectfully ask the Area Chair to consider our clarifications and the newly provided evidence when making the final recommendation.

Sincerely,
The Authors

---

### Meta-Review · Area_Chair_GiX7 · 2026-01-04

**Summary:**

The reviewers initially pushed back on this paper because they felt it lacked a thorough comparison with the latest research and didn't clearly justify its complex technical choices. Specifically, they were looking for a more detailed literature survey, data on how long the model takes to run (inference time), and a better explanation for why the authors used reinforcement learning (RL) instead of simpler fine-tuning methods. Despite these gaps, everyone agreed that the paper tackles a massive problem: AI image detectors usually fall apart when an image is slightly compressed or edited. The authors' proposed method, AIGID-RFT, shows a lot of promise in staying accurate even when images are "roughed up" by JPEG compression. However, the computational speed is one of the concerns, as it only processes 0.284 frames per second (FPS), whereas other methods could run as fast as 15 FPS. Meanwhile, the reviewers also raised concerns about replacing the Qwen 2.5 with other models, restricting the adaptability of the proposed method. Additionally, several reviewers also pointed out that the literature reviews are not thorough, and the insights into leveraging RL in generated image detection are unclear. Hence, the AC recommends rejection.

**Reviewer Concerns:**

Reviewer tuqj pointed out that the model still struggles with face-swap (Deepfake) images. The authors explained this is likely because their training data mostly featured objects like cars and cats rather than human faces. While the explanation makes sense, it confirms the model isn't a "silver bullet" for all types of fake media yet.

Reviewer hXs7 requested the authors to experiment with recent foundation models, which is reasonble, the authors do not provide such justification.

Several reviewers also raised concerns regarding the novelty, challenging the authors that the proposed method is simple combination of RL and features from foundation models, lacking of insights.

**Reviewer Scores:**

Reviewer dDMg: 6

Reviewer tuqj: 2

Reviewer hXs7: 2

Reviewer pa3X: 8

The reivewer tuqj and hXs7 are unlikely to increase their ratings, as the major concerns about the novelty and deep-fake detection are not addressed. The AC is uncertain about the reviewer pa3X, as the latency requested by the reviewer is very much lower than the previous methods.

---

### Decision · Program_Chairs · 2026-01-26

Reject